# Impact of Circadian Clock *PER2* Gene Overexpression on Rumen Epithelial Cell Dynamics and VFA Transport Protein Expression

**DOI:** 10.3390/ijms252212428

**Published:** 2024-11-19

**Authors:** Rahmat Ali, Yongkang Zhen, Xi Zanna, Jiaqi Lin, Chong Zhang, Jianjun Ma, Yuhong Zhong, Hosameldeen Mohamed Husien, Ahmad A. Saleh, Mengzhi Wang

**Affiliations:** 1Laboratory of Metabolic Manipulation of Herbivorous Animal Nutrition, College of Animal Science and Technology, Yangzhou University, Yangzhou 225009, China; dh20022@stu.yzu.edu.cn (R.A.); 008643@yzu.edu.cn (H.M.H.); 2College of Animal Science & Technology, Yangzhou University, Yangzhou 225009, China; elemlak1339@gmail.com

**Keywords:** *PER2* gene, rumen epithelium, the circadian clock, circadian rhythm, volatile fatty acid transporter protein

## Abstract

The circadian gene *PER2* is recognized for its regulatory effects on cell proliferation and lipid metabolism across various non-ruminant cells. This study investigates the influence of *PER2* gene overexpression on goat rumen epithelial cells using a constructed pcDNA3.1-*PER2* plasmid, assessing its impact on circadian gene expression, cell proliferation, and mRNA levels of short-chain fatty acid (SCFA) transporters, alongside genes related to lipid metabolism, cell proliferation, and apoptosis. Rumen epithelial cells were obtained every four hours from healthy dairy goats (n = 3; aged 1.5 years; average weight 45.34 ± 4.28 kg), cultured for 48 h in vitro, and segregated into control (pcDNA3.1) and overexpressed (pcDNA3.1-*PER2*) groups, each with four biological replicates. The study examined the potential connection between circadian rhythms and nutrient assimilation in ruminant, including cell proliferation, apoptosis, cell cycle dynamics, and antioxidant activity and the expression of circadian-related genes, VFA transporter genes and regulatory factors. The introduction of the pcDNA3.1-*PER2* plasmid drastically elevated *PER2* expression levels by 3471.48-fold compared to controls (*p* < 0.01), confirming effective overexpression. *PER2* overexpression resulted in a significant increase in apoptosis rates (*p* < 0.05) and a notable reduction in cell proliferation at 24 and 48 h post-transfection (*p* < 0.05), illustrating an inhibitory effect on rumen epithelial cell growth. *PER2* elevation significantly boosted the expression of *CCND1*, *WEE1*, *p21*, and *p16* (*p* < 0.05) while diminishing *CDK4* expression (*p* < 0.05). While the general expression of intracellular inflammation genes remained stable, *TNF-α* expression notably increased. Antioxidant marker levels (SOD, MDA, GSH-Px, CAT, and T-AOC) exhibited no significant change, suggesting no oxidative damage due to *PER2* overexpression. Furthermore, *PER2* overexpression significantly downregulated *AE2*, *NHE1*, *MCT1*, and *MCT4* mRNA expressions while upregulating *PAT1* and *VH+ ATPase*. These results suggest that *PER2* overexpression impairs cell proliferation, enhances apoptosis, and modulates VFA transporter-related factors in the rumen epithelium. This study implies that the *PER2* gene may regulate VFA absorption through modulation of VFA transporters in rumen epithelial cells, necessitating further research into its specific regulatory mechanisms.

## 1. Introduction

The circadian clock system governs the expression of clock-controlled genes (CCGs), which influence several physiological functions [1]. This regulatory framework is crucial for maintaining homeostasis and optimizing the timing and coordination of biological activities according to the 24 h circadian rhythm [2]. The circadian cycle is driven by complex feedback loops that link transcription and translation. Core clock genes that produce rhythmic patterns of gene expression, such as *CLOCK*, *BMAL1*, *PER*, and *CRY*, are involved in these loops [3]. Two essential elements of the positive limb of the circadian clock are *BMAL1* and *CLOCK*. In order to start the transcription of target genes, including *Period (PER)* and *Cryptochrome (CRY)*, they form a heterodimer that binds to E-box motifs in their promoters [4,5].

The rumen is a vital organ in ruminants, serving as the primary site for microbial fermentation of ingested feed. It plays a critical role in nutrient absorption, converting complex plant materials into volatile fatty acids, amino acids, and other essential nutrients that are pivotal for the animal’s energy metabolism and overall health [6,7]. Understanding rumen physiology is crucial not only for optimizing ruminant nutrition and enhancing productivity but also for maintaining animal health and welfare. Recent studies have suggested that circadian rhythms, which are endogenously generated near-24 h cycles, profoundly influence various physiological processes, including metabolism, digestion, and immune function [8,9]. The *PER2* gene, a core component of the circadian clock, regulates these rhythms and may significantly affect rumen functionality by modulating gene expression linked to nutrient absorption and cellular dynamics. Therefore, examining the circadian regulation of the rumen through the lens of PER2 function provides valuable insights into how temporal organization impacts ruminant physiology, with potential implications for improving animal health and productivity [10,11].

The PER and CRY proteins, which are essential for the circadian feedback loop, are expressed when the BMAL1-CLOCK complex is activated. Following production, these proteins return to the nucleus to inhibit BMAL1-CLOCK activity, which controls the expression of these proteins [12,13]. While PER3 may affect rhythm amplitude and resilience [14], *PER1* and *PER2* are crucial for preserving the clock’s precision and stability [15,16]. *CRY* proteins, including *CRY1* and *CRY3*, are indispensable for the inhibition of *BMAL1-CLOCK* and help synchronize the circadian clock with light–dark cycles via photic entrainment [4,17].

The key circadian clock gene, *PER2*, significantly influences cell proliferation and lipid metabolism. Previous research on non-ruminant cells indicates that this circadian clock gene affects multiple cellular activities, including cell division, apoptosis, lipid metabolism, and mRNA levels associated with amino acid transporters [18,19,20,21]. Moreover, research indicates that circadian clock genes in non-ruminant gastrointestinal systems significantly regulate macronutrient absorption and transport [22]. Casey and Plaut [23] suggested, based on rodent models, that circadian clocks are essential for maintaining lactation homeostasis by affecting hormone profiles, such as prolactin, and altering metabolic pathways [23].

Most studies on ruminants have focused on the function of circadian clock genes in metabolic organs such as the liver, adipose tissue, and mammary gland, as well as their interaction with nuclear receptors, specifically peroxisome proliferator-activated receptors (PPARs), which are essential for lipid metabolism [24,25]. In mammary cells, studies indicate that *PER2* partly controls α-casein protein production [26]. Circadian clock components’ presence and physiological functions in rapidly proliferating tissues, such as the ruminal epithelium, remain uncertain. This knowledge gap underscores the demand for more studies to explore the expression and functional importance of circadian clock genes in rumen and their potential impact on ruminal physiology and overall animal health.

Nutrient transporters involved in the absorption of glucose, peptides, and lipids exhibit different diurnal rhythms in their mRNA expression levels, according to studies conducted in non-ruminants [27,28]. Additionally, studies using rodent models have shown that the circadian clock affects the mRNA levels of ion transporters in the intestines, particularly SLC4A1 (AE1) and SLC9A3 (NHE3), with peak expression taking place at the beginning of the night phase [29]. Additionally, research in controlled settings has shown that BMAL1, a crucial gene linked to circadian rhythm, can influence glucose absorption in Caco-2 cells by modifying the translation of SLC5A1 (SGLT1) [30,31,32].

Considering these findings, we hypothesized that ruminal epithelial cells (RECs) may also express components of the circadian clock and that PER2 may be essential for controlling cellular proliferation and nutrient transporter mRNA expression levels in these cells.

This hypothesis aims to investigate the potential connection between circadian rhythms and nutrient assimilation in ruminants, building upon the foundational knowledge derived from non-ruminant models.

The interaction between these genes is intricate and precisely regulated, emphasizing the sophisticated nature of circadian regulation [33]. This study seeks to further explore this complexity by examining the effects of *PER2* overexpression on rumen epithelial cell proliferation and VFA absorption protein expression.

## 2. Results 

### 2.1. Detection of PER2 Overexpression Efficiency

The *PER2* gene showed a relative expression level of 4061.63 after 48 h of transfection with the pcDNA3.1-*PER2* plasmid (Figure 1). The overexpression group exhibited a 3471.48-fold increase in *PER2* expression compared to the control group, representing a statistically significant difference (*p* < 0.01). These findings confirm the successful overexpression of the *PER2* gene in goats, establishing a solid foundation for subsequent experiments.

### 2.2. Effect of PER2 Overexpression on Cell Proliferative Activity and Apoptosis of Goat Rumen Epithelial Cells

Flow cytometry was employed to assess apoptosis in goat rumen epithelial cells 48 h after transfection with the pcDNA3.1-*PER2* plasmid (Figure 2A). The results showed a significant rise in apoptosis levels among the *PER2* overexpression set when compared to the control set (*p* < 0.05) (Figure 2B). Furthermore, the CCK8 proliferation assay evaluated cell viability at 24 and 48 h post-transfection. The findings indicated a markedly diminished cellular viability within the PER2 overexpression cohort in contrast to the control cohort at both temporal measurements (*p* < 0.05) (Figure 2C). These results suggest that *PER2* overexpression reduces the survival of goat rumen epithelial cells.

According to Table 1, the intensified expression of the PER2 gene has considerably improved the levels of pro-apoptotic genes P21, BAX, and CASPASE6 (*p* < 0.05). In contrast, the anti-apoptotic gene BCL2 showed significantly reduced expression (*p* < 0.05). However, other apoptosis-related genes, including CASPASE3, CASPASE7, CASPASE8, CASPASE9, and *p53*, exhibited no significant alterations in expression levels (*p* > 0.05).

### 2.3. Effect of PER2 Overexpression on Cell Cycle Gene Expression in Goat Rumen Epithelial Cells

Overexpression of the PER2 gene led to a notable increase in the expression levels of CCND1, WEE1, p21, and *p16* when compared to the control group (*p* < 0.05), as shown in (Table 2). Furthermore, a significant decrease in *CDK4* gene expression was observed (*p* < 0.05). The enhanced *PER2* gene expression resulted in substantially higher levels of *CCND1*, *WEE1*, *p21*, and *p16* relative to the control set (*p* < 0.05), as illustrated in Table 2. Moreover, a considerable reduction in *CDK4* gene expression was noted (*p* < 0.05).

### 2.4. Effect of PER2 Overexpression on Inflammatory Gene Expression in Goat Rumen Epithelial Cells

As shown in Figure 3 and Table 3, overexpression of the *PER2* gene did not lead to significant changes in the expression of intracellular inflammatory genes (*p* > 0.05), with the exception of *TNF-α*, which increased significantly.

### 2.5. Effect of PER2 Gene Overexpression on Rumen Epithelial Cell Membrane Damage in Goats

LDH release was measured to assess the impact of *PER2* gene overexpression on membrane damage in goat rumen epithelial cells. As shown in Figure 3, there was no significant difference in LDH release between the control group (pcDNA3.1) and the overexpression group (pcDNA3.1-*PER2*) (*p* > 0.05).

### 2.6. Effect of PER2 Gene Overexpression on ROS in Goat Rumen Epithelial Cells

As shown in Figure 4, the DEFH-DA probe was used to measure ROS levels in goat rumen epithelial cells. The results indicated no significant change in ROS fluorescence intensity in the *PER2* overexpression group compared to the control group. This suggests that *PER2* gene overexpression did not cause oxidative damage to the rumen epithelial cells.

### 2.7. Effect of PER2 Gene Overexpression on Antioxidants in Goat Rumen Epithelial Cells

As shown in Figure 5, the antioxidant performance of goat rumen epithelial cells was evaluated using a detection kit. The results revealed no significant changes in the levels of intracellular SOD, MDA, GSH-PX, CAT, or T-AOC in the *PER2* overexpression group compared to the control group. These findings suggest that *PER2* gene overexpression did not induce oxidative damage in goat rumen epithelial cells.

### 2.8. Effect of PER2 Gene Overexpression on VFA Transporters and Related Factors in Rumen Epithelial Cells

The mRNA expression levels of VFA transporters and related factors are presented in Table 4. Overexpression of the *PER2* gene resulted in significantly reduced relative expression levels of the AE2, NHE1, MCT1, and MCT4 genes when compared to the control group (*p* < 0.01). In contrast, the PAT1 and VH+ ATPase genes exhibited significantly higher relative expression levels in the overexpression group (*p* < 0.05). While NHE3 and NA/K ATPase genes showed increased relative expression levels in the overexpression group, these differences were not statistically significant (*p* > 0.05).

## 3. Discussion

Before exploring the effects of *PER2* overexpression on goat rumen epithelial cells, it is crucial to understand its broader biological role across various species. In pigs, sheep, and cattle, *PER2* is essential for regulating circadian rhythms, influencing significant physiological and metabolic processes, which provides valuable insights for the current study.

In pigs, circadian regulation by *PER2* is vital for maintaining metabolic balance. The interaction of *PER2* with other clock genes such as *BMAL1* and *CRY2* affects lipid metabolism, which in turn influences growth and meat quality [34]. Zhou et al. [35] demonstrated that diurnal variations in fatty acid metabolism highlight the significance of timing in feed administration. Alterations in *PER2* expression have been linked to changed lipid metabolism, suggesting potential improvements in growth performance and a reduction in liver fat accumulation through its manipulation [36]. Furthermore, synchronizing feeding schedules with circadian rhythms boosts muscle quality and milk lipid profiles, underscoring *PER2’*s role in aligning metabolic activities with environmental cues for optimum productivity [37,38].

In sheep, *PER2* expression is closely associated with seasonal reproductive cycles driven by photoperiod changes [39,40]. This synchronization ensures that reproductive activities occur under favorable conditions. *PER2* cycles, along with other clock genes, facilitate fertility optimization and adapt metabolic functions to seasonal changes [41,42]. Under extended daylight, *PER2* expression lengthens, correlating with shifts in reproductive hormone levels, such as kisspeptin [41]. Understanding *PER2* modulation can enhance breeding strategies aligned with natural photoperiods, improving reproductive outcomes [43].

In cattle, circadian rhythms significantly impact processes like reproduction and lactation, with *PER2* and its interaction with other major clock genes playing a key role [44]. Diurnal changes in luteinizing hormone levels during the estrous cycle suggest a circadian influence linked to *PER2* [45]. *PER2* regulates milk production by influencing the synthesis of milk proteins and metabolic genes during lactation [25,46]. Photoperiod adjustments, like increased light exposure, enhance milk yields partially due to altered *PER2* expression, which improves metabolic processes [47,48,49]. Modulating *PER2* in bovine systems highlights its importance in boosting dairy and reproductive performance, offering insights into management practices that leverage circadian biology for increased productivity [50,51].

Overall, *PER2* is a crucial component within the circadian system which is intricately involved in regulating metabolic and reproductive functions across various species. Its synchronization with environmental stimuli presents practical applications in agriculture, promising enhancements in productivity, and resource efficiency by aligning management practices with innate biological rhythms.

### 3.1. Effect of PER2 Gene Overexpression on the Proliferative Activity and Apoptosis Level of Goat Rumen Epithelial Cells

The circadian rhythm regulates many functions related to the normal physiological metabolism of organisms, leading to periodic changes within a 24 h cycle. Current research demonstrates that clock genes influence various physiological activities, including cell proliferation and apoptosis [52,53,54,55]. In non-ruminant models, overexpression of *PER1* has been shown to inhibit cell growth [56,57,58]. *PER2*, a key core clock gene, acts as an important negative regulator in the circadian feedback loop, maintaining both the circadian rhythm of cells and a normal cell cycle [59,60,61,62]. Research has demonstrated that PER2 can regulate cell proliferation and apoptosis [19,63,64]. Several other investigations revealed that the boosted expression of the *PER2* gene fosters autophagy and apoptosis, while concurrently restraining OSCC cell growth. Moreover, *PER2* overexpression exhibits a substantial growth-inhibitory effect on mouse tumor cells, playing a vital role in tumor suppression by triggering apoptotic cell death [54,65].

In studies involving rumen epithelial cells and mammary epithelial cells, silencing the *PER2* gene has been shown to promote cell proliferation and inhibit apoptosis [66,67], while overexpression of *PER2* exacerbates apoptosis [54,68,69]. Gao et al. [32] found that treating rumen epithelial cells with 15 mM sodium butyrate enhanced *PER2* gene expression, which resulted in a surplus of *PER2* production. This surplus was linked to decreased cell proliferation and gene alterations related to the cell cycle, thus impacting cell division. In this investigation, we noted a pronounced elevation in the levels of apoptosis in goat rumen epithelial cells after the overexpression of *PER2*. BCL2 family proteins are key mitochondrial responses to apoptotic signaling regulators, typically inhibiting cell apoptosis. In contrast, BAX is a pro-apoptotic member that promotes apoptotic cell death. The interplay between anti-apoptotic and pro-apoptotic molecules determines how cells respond to apoptotic signals [70,71]. In this investigation, mRNA levels of apoptosis-related factors were evaluated using qRT-PCR. The results showed that in the control group treated with void plasmids, increased *PER2* concentrations significantly enhanced the expression of pro-apoptotic genes, such as *CASP6* and *BAX*, while simultaneously causing a substantial decrease in the expression of the anti-apoptotic gene *BCL2*.

*p21* acts as an inhibitor of cyclin-dependent kinases, inducing cell cycle arrest during the G phase by blocking CDK activity and relevant DNA replication factors. This leads to the cessation of cell division and proliferation, thereby inhibiting cellular growth. The expression of the *p21* gene is primarily regulated by *p*53, which induces *p21* expression in response to DNA damage [72,73,74]. In this trial, consistent with previous reports, overexpression of *PER2* increased the mRNA levels of *p53* and *p21* in goat rumen epithelial cells [75]. Additionally, cell viability in the overexpression group was significantly lower than in the control group, corroborating the aforementioned results and further confirming that PER2 may modulate cell proliferation and apoptosis by influencing the expression of various apoptosis-related factors.

*PER2*’s involvement in cellular mechanisms extends beyond its circadian regulatory functions, notably through its interaction with the *p53* pathway. *PER2* has been recognized to modulate the stability and transcriptional activity of *p53*, a pivotal tumor suppressor involved in cell cycle arrest and apoptosis. The overexpression of *PER2* can enhance *p53* stabilization, thereby augmenting the transcription of downstream targets such as *p21*, which contributes to G1 cell cycle arrest [76,77,78]. This interaction suggests a mechanism by which PER2 may influence apoptosis through the upregulation of pro-apoptotic genes and downregulation of anti-apoptotic genes. Additionally, this could explain *PER2*’s inhibitory effects on cell proliferation, as increased p53 activity hampers cell cycle progression and promotes apoptotic pathways. Understanding this *PER2*-p53 axis provides significant insights into how circadian clock genes might exert control over cell fate decisions, especially in rumen epithelial cells under varying physiological conditions.

### 3.2. Effect of PER2 Gene Overexpression on Cell Cycle Gene Expression in Goat Rumen Epithelial Cells

The normal activity of the cell cycle is tightly regulated by various cell cycle genes and proteins, with cyclin-dependent kinases (CDKs) and cyclins playing central roles. Recent scholarly investigations have elucidated a robust correlation between the cellular cycle and the circadian rhythm, emphasizing that numerous genes implicated in the cellular cycle are concurrently regulated by genes governing the circadian clock. For example, it has been reported that CCND and WEE1 are directly regulated by clock genes [15,73]. The *PER2* gene, an essential element of the circadian rhythm, contributes substantially to this mechanism by modulating an array of downstream genes associated with the cell cycle. Abnormal expression of the *PER2* gene can disrupt the cell cycle, leading to imbalanced expression of cell cycle-related genes, which in turn causes disturbances in the cell cycle and an imbalance between cell growth and apoptosis [55,79].

Cell cycle-related factors are essential for the proper functioning of the cell cycle, ensuring a balance between cell proliferation and apoptosis. The cell cycle is regulated by a molecular network that includes cyclins, CDKs, and CDK inhibitors (CKIs). CDKs are at the core of this regulatory system and are modulated by both cyclins and CKIs. The main cyclins are CCNA2, CCNB1, CCND1, and CCNE1, and they work alongside vital CDKs like CDK1, CDK2, CDK4, and CDK6 for the advancement of the cell cycle. CKIs, particularly *p16* and *p21*, play a crucial role in this regulatory network [80,81]. During each part of the cell cycle, designated cyclins bind to their assigned CDKs to create cyclin/CDK complexes, leading to the activation of CDKs and supporting the organized flow of the cell cycle. Conversely, CKIs can inhibit cell cycle progression by binding to specific CDKs or cyclin/CDK complexes, thereby inhibiting CDK activity [80,82].

This study demonstrated that enhancing *PER2* gene expression in rumen epithelial cells led to a notable increase in *CCND1*, *WEE1*, *p21*, and *p16* levels, while *CDK4* gene expression decreased significantly. These findings align with previous research on *PER2* gene interference in human oral squamous cell carcinoma cells [19]. Specifically, *p21* induces cell cycle arrest in the G phase by inhibiting CDKs and DNA replication factors, resulting in the cessation of cell division and proliferation, thus impeding cell growth [83,84]. This suggests that PER2 plays a crucial role in regulating the cyclin–CDK-CKI cell cycle regulatory network.

### 3.3. Effect of PER2 Gene Overexpression on Anti-Inflammatory Factors and Antioxidant Properties of Goat Rumen Epithelial Cells

Multiple studies have shown that core clock genes and inflammatory factors in cells exhibit a circadian rhythm [85,86,87]. Toll-like receptors (*TLRs*) are crucial for innate immune recognition and cell activation in response to antigens, thus facilitating immunoinflammatory responses. *TLR2* and *TLR4* are particularly important among *TLRs*. Research has shown that *TLR2* and *TLR4* can trigger NF-κB translocation by binding to intestinal LPS, subsequently inducing the expression of inflammatory cytokines such as *IL-1*, *IL-6*, *IL-8*, *TNF-α*, and *INF-γ* [88].

Further investigations have shown that knocking out the *Per2* gene in rat RNK16 cells affects the expression of immune modulators like *INF-γ* and *TNF-α* to some extent. Similarly, studies involving LPS-stimulated *BV2* cells from *Per2* knockout mice revealed substantial increases in TNF-α and IL-6 levels within the cell culture medium [89]. Additionally, earlier research revealed a significant reduction in *TLR2* expression in the colon of *Per2* gene knockout mice [90]. Combined with the result of the previous findings, this study demonstrated that overexpression of the *PER2* gene in goat rumen epithelial cells led to a significant increase in *TNF-α* expression, while there was no significant impact on anti-inflammatory factors or antioxidant activity. This could be attributed to the possibility that inflammation is linked to immune responses involving certain gut microbiota interacting with Toll-like receptors, a process that in vitro cell assays may not accurately replicate. Another potential explanation is that the increase in *TNF-α* expression might promote apoptosis. Future studies should investigate the specific mechanisms underlying these observations.

### 3.4. Effect of PER2 Gene Overexpression on VFA Transporters and Related Factors in Goat Rumen Epithelial Cells

Circadian clock genes are essential in modulating the uptake and translocation of macronutrients within the gastrointestinal system [91,92]. Recent studies have increasingly shown that the *PER2* gene can influence lipid metabolism [20,93] and affect the mRNA abundance of amino acid transporters [39,94]. The AE2 protein, located in the epithelial cell membrane of the rumen, regulates the exchange of volatile fatty acids (VFAs) and bicarbonate (HCO_3_^−^), playing a crucial role in maintaining internal balance. *AE2*, alongside chloride ions (Cl^−^), contributes to VFA absorption and transport [95].

The Na^+^/H^+^ exchanger (NHE) family, especially *NHE1*, includes functions that affect how cells grow and die in a programmed manner, sustain cellular volume balance, and regulate pH levels inside the cell along with ionic transport. *NHE1* activation has been shown to promote rapid progression through the G1 phase of the cell cycle; conversely, its inhibition can cause significant delays in the S phase, potentially arresting cell division [96]. *MCT1* and *MCT4*, the transporters in question, are essential for moving short-chain fatty acids (SCFAs) through the epithelial cells of the rumen, helping with their transluminal transfer alongside ketone bodies and lactate [97]. Essentially, research demonstrates that the overexpression of *PER2* driven by butyrate results in diminished mRNA levels of *MCT1* and *MCT4*, while concurrently boosting the mRNA levels of V-H+-ATPase [32].

This research revealed that heightened levels of *PER2* in rumen epithelial cells led to a pronounced decrease in *AE2*, *NHE1*, *MCT1*, and *MCT4* expressions, whereas *PAT1* and V-H+-ATPase expressions demonstrated a noteworthy increase. This might be due to *PER2*’s regulatory effect on MCTs via the *p53* pathway. The *p53* protein acts as an important transcription factor involved in various cellular signaling mechanisms, with its expression influenced by the *PER2* protein. It has been reported that *PER2* overexpression can enhance *p53* expression in vitro [54,98]. High concentrations of butyrate can further upregulate *p53* expression, inducing apoptosis in rumen epithelial cells and reducing cell viability [99]. Conversely, *P53* has been shown to inhibit *MCT1* expression, leading to reduced lactate excretion [100,101].

The investigation we conducted featured a quantitative assessment of gene expression via qRT-PCR, suggesting that the decline in gene expression did not align with reduced cell density. Therefore, it can be speculated that *PER2* may regulate MCTs by influencing cell proliferation, growth, development, and the cell cycle in rumen epithelium, ultimately affecting VFA absorption and metabolism. Taken together with earlier findings, it appears that *PER2* can modulate the transcription of VFA transport-related factors in rumen epithelial cells, thereby playing a role in regulating SCFA absorption and lipid metabolism.

Understanding the role of the *PER2* gene in the rumen presents a significant opportunity to influence livestock management, dietary formulations, and enhance ruminant productivity. PER2 is integral to the regulation of circadian rhythms, which in turn affect metabolic processes. By elucidating how *PER2* functions within the rumen’s microbial ecosystem, we can tailor feeding schedules and dietary ingredients to align better with the animals’ natural metabolic cycles. This alignment can improve digestion efficiency and nutrient absorption, potentially leading to faster growth rates and higher milk yields. Additionally, understanding *PER2* could help in developing precision feeding strategies that enhance animal health and reduce waste, thereby optimizing overall productivity and sustainability in ruminant farming. [23,102,103,104,105,106,107].

Numerous studies have delved into the roles of circadian clock genes, such as CLOCK, BMAL1, PER, and CRY, in the liver and mammary glands, highlighting their complex involvement in metabolic pathways and physiological regulation [102,103]. In the liver, which serves as a crucial metabolic center, these genes coordinate essential processes including glucose homeostasis, lipid metabolism, and detoxification. Specifically, the PER2 gene, part of the period family, plays a pivotal role in maintaining the feedback loops that regulate these rhythms. It functions as a negative regulator, interacting with other clock proteins to modulate the timing of gene expression involved in metabolic pathways [104,105,106].

Similarly, in the mammary glands, circadian clock genes, including *PER2*, regulate lactation cycles by controlling the timing of milk synthesis-related gene expression. This regulation directly affects milk quality and production yield, ensuring synchronization with the organism’s biological rhythms. The *PER2* protein acts as a crucial component in this regulatory network, influencing the transcription of genes crucial for sustaining lactation cycles within optimal periods [23,107].

Despite these advancements, there is a notable gap in our understanding of the expression and functional impact of circadian clock genes, particularly *PER2*, within the rumen. The rumen is essential for microbial fermentation and nutrient processing in ruminants. Here, circadian rhythms might play a crucial role, with *PER2* potentially influencing the temporal organization of microbial enzyme activity, thereby affecting feed conversion and nutrient absorption [105,108]. Investigating how *PER2* and other circadian genes function in the rumen could lead to innovative strategies to improve digestive efficiency, enhance animal health, and boost production outputs. Therefore, research focused on uncovering circadian regulation in the rumen is a pivotal and largely unexplored field with promising potential benefits spanning both scientific inquiry and industrial application [109,110]. Investing in this research area could yield significant insights and advancements in agricultural practices and livestock management, ultimately contributing to better economic and health outcomes for the industry [111].

## 4. Materials and Methods

### 4.1. Statement of Ethical Approval

The animal studies in this work followed the ethical norms and protocols approved by the Animal Care and Use Committee (202203-512 Approval) at Yangzhou University, Jiangsu, China.

### 4.2. Experimental Design

This study aimed to investigate the effects of overexpression of the *PER2* gene in rumen epithelial cells. To increase *PER2* expression, we employed gene overexpression strategies. The cells were separated into a control group (pcDNA3.1) and an overexpression group (pcDNA3.1-PER2), each with four biological replicates.

After culturing the cells, samples were collected at 24 and 48 h to examine various parameters: cell proliferation, apoptosis, and cell cycle progression. Additionally, we investigated alterations in inflammatory indicators, antioxidant activity, PER2 expression, and proteins involved in volatile fatty acid absorption. Our goal was to learn how the PER2 protein affects the uptake of volatile fatty acids in rumen epithelial cells, particularly in terms of circadian rhythm control.

### 4.3. Rumen Epithelial Tissue, Cell Total RNA Extraction, and qRT-PCR

The initial digestion of primary cells involved a 3 min treatment with a 0.25% trypsin and 0.02% Na-EDTA solution, followed by a 5 min digestion using fresh trypsin–EDTA solution. The resulting cells were subsequently seeded at 2 × 10⁶ cells/mL, then cultured in a growth medium for passaging. Third-generation cells were obtained within two weeks and progressively cryopreserved at −80 °C in a mixture of 10% DMSO, 50% fetal bovine serum, and 40% DMEM/F12. These stored cells were later revived and cultured to the sixth generation for experimental use, aged under 4 months at the start of the initial experiment.

The rumen epithelial cells (RECs) were isolated from three healthy Guanzhong dairy goats (all females, 1.5 years of age, approximately 45.34 ± 4.28 kg body weight, which is roughly equivalent to a 21-year-old human) using a modified serial trypsin digestion method [112]. The extracted RECs were cultured in a medium composed of DMEM and Nutrient Mixture F-12, enriched with 5% fetal bovine serum, antibiotics (200 U/mL penicillin, 0.2 mg/mL streptomycin, 0.1 mg/mL gentamicin), 5 μg/mL amphotericin B, 1% insulin–transferrin–selenium solution, and 10 ng/mL epidermal growth factor. The culture medium was changed every two days. To maintain REC purity and prevent fibroblast contamination, the cells underwent repeated trypsin digestion and differential attachment processes. Microscopic evaluation confirmed the absence of other cell types. Before the experiments, immunofluorescence staining was used to confirm that the cells were epithelial by detecting the cytokeratin 18 protein. The functionality of the isolated REC was assessed by culturing them with either 0 mM (control) or 15 mM sodium butyrate for 24 h. The treated cells exhibited higher BHB concentrations, demonstrating their ability to metabolize butyrate, a crucial function of REC [113].

The rumen epithelium consists of four distinct layers, with ketogenic enzymes predominantly situated in the mitochondria of stratum basale cells. This observation led researchers to infer that isolated RECs originated from this particular layer. An initial 3 min exposure to a solution comprising 0.25% trypsin and 0.02% Na-EDTA was part of the isolation technique. This was followed by a 5 min treatment with a new trypsin–EDTA solution. Following harvest, the cells were planted in growth media for passage at a density of 2 × 10⁶ cells/mL. After being collected within two weeks, third-generation cells were gradually frozen and stored at −80 °C in a solution that contained 40% DMEM/F12, 50% fetal bovine serum, and 10% DMSO. These cryopreserved cells were later revived and cultured to the sixth generation for experimental purposes, ensuring they were no older than 4 months at the onset of the first experiment.

RNA extraction was conducted using the Trizol method, following a protocol previously established by Wang et al. [114]. To synthesize cDNA, 1 μg of RNA was reverse transcribed using the Tiangen FastQuantc DNA First Strand Synthesis Kit (KR106, Tiangen, Beijing, China) according to the manufacturer’s instructions. Oligo 6 software was utilized to design primers for target genes and internal controls (ACTB and GAPDH), which were subsequently produced by Invitrogen Trading Co. Ltd. (Shanghai, China). Real-time PCR was performed using an Applied Biosystems 7500 Real-Time PCR System (Thermo Fisher Scientific, Waltham, MA, USA) with SuperReal PreMix Plus (SYBR Green) from TIANGEN Biotech Co. Ltd., Beijing, China (FP215). To ensure reliability, each reaction was performed in triplicate. Additional information regarding the RT-PCR procedure can be found in our earlier publication [114]. The specificity of primers was verified through dissociation curve analysis during RT-PCR and further confirmed by agarose gel electrophoresis of PCR products.

### 4.4. CDS Sequence Cloning

The goat *PER2* gene’s CDS was cloned by designing primers based on the full mRNA sequence available on GenBank. The primers used were *PER2*-F (forward): CCCAAGCTTAGAGCCAGCATGGACGGC and *PER2*-R (reverse): CCGGAATTCCTAGCGGACGTCGCTGGC. The forward primer included a *Hind III* digestion site, while the reverse primer included an *EcoRI* enzyme cleavage site and protective bases. Nanjing Qingke Biotechnology Co., Ltd., Nanjing, China, synthesized these primers. The plasmid structure is shown in (Figure 6a).

The coding sequence (CDS) of the goat *PER2* gene was amplified using the cDNA synthesized in the previous step as the template. The PCR reaction mixture (50 μL total) contained PrimeSTAR GXL DNA Polymerase (1 μL, 1.25 U/μL), 5X PrimeSTAR GXL Buffer (10 μL), dNTP Mixture (4 μL, 2.5 mM each), upstream and downstream primers (1.5 μL each), cDNA template (5 μL), and sterilized water to reach the final volume.

PCR conditions included initial pre-denaturation (98 °C, 5 min), followed by 35 cycles of 98 °C for 10 s, 60 °C for 15 s, and 68 °C for 30 s. A final extension (68 °C, 10 min) was performed before termination and storage at 4 °C. The PCR product underwent 1% agarose gel electrophoresis to verify the amplified fragment’s specificity and size.

After confirming successful amplification via gel electrophoresis, the target DNA fragment was cut out from the gel and purified. The *PER2* CDS amplification is shown in Figure 6b. The purified product was then ligated into the pMD19-T vector following the addition of an A-tail. A 10 μL portion of the ligation mixture was used to transform *E. coli* DH5α competent cells. The transformed cells were plated onto selection plates containing 1% ampicillin and incubated overnight at 37 °C.

Following incubation, bacterial colonies were picked, cultured in a shaking incubator, and subjected to PCR screening to identify positive clones. The selected positive clones were sent to Nanjing Qingke Biotechnology Co., Ltd. for sequencing. Based on the sequencing results, positive clones were further verified through digestion and confirmed to have the correct sequence. The successfully constructed T-*PER2* plasmid was preserved by mixing the bacterial culture with sterilized 50% glycerol at a 2:3 volume ratio for long-term storage.

### 4.5. Overexpression Vector (pcDNA3.1-PER2) Construction

To construct the pcDNA3.1-*PER2* overexpression vector, the previously constructed T-*PER2* plasmid and the pcDNA3.1 vector were digested separately using the restriction enzymes *Hind III* and *EcoRI* (Figure 6c). Following digestion, the desired fragments were extracted from the gel and purified. These fragments were then joined using T4 DNA ligase to form the recombinant plasmid. *E. coli* DH5α competent cells were transformed with the ligation product and cultured on ampicillin-containing screening plates. Colonies were selected based on antibiotic resistance, and positive clones were sequenced to verify correct *PER2* gene insertion. Following confirmation by sequencing, the positive clones were used to extract endotoxin-free plasmids. These plasmids were then subjected to enzymatic digestion to verify the successful construction of the pcDNA3.1-*PER2* plasmid.

Finally, the successfully constructed plasmid and the corresponding bacterial culture were preserved. The bacterial culture was mixed with sterilized 50% glycerol and stored for long-term use.

### 4.6. Transient Transfection of Cells

Transient transfection was performed using Lipofectamine™ 2000 transfection reagent (Thermo-fisher, Thousand Oaks, California, USA). Depending on experimental requirements, cells were seeded in 6-well, 24-well, or 96-well plates one day before transfection. When the cells reached approximately 70% to 80% confluency, they were washed with PBS, and an appropriate amount of culture medium or Opti-MEM medium was added. Endotoxin-free plasmids were diluted in a serum-free medium at a ratio of 1:1 to 1:3 (μg plasmid: μL Lipofectamine). The plasmid and Lipofectamine reagent were diluted and mixed in a 1:1 volume ratio before incubating at room temperature for 20 min to produce DNA–lipid complexes. These complexes were added to cells in well plates, with four replicates per experimental group. After 3 to 6 h of incubation, the transfection medium was replaced with a full-growth medium. The cells were then grown for 24 to 48 h before being used in further tests.

### 4.7. Cell Proliferation Viability Detection

The CCK8 kit was utilized to evaluate cell proliferation. Cells were distributed in 96-well plates, with 100 μL per well. The experiment had two groups, control (pcDNA3.1) and overexpression (pcDNA3.1-PER2), each with four replicate wells. After 24–48 h of the incubation process, each well was filled with 10 μL of CCK8. The plates were gently stirred to achieve even reagent dispersion and then incubated in a cell culture incubator for two hours. To assess cell growth rates, the absorbance at 450 nm was measured using a microplate reader following incubation.

### 4.8. Apoptosis Detection

Cells were kept in 6-well plates and segregated into two separate experimental classifications: the control classification (pcDNA3.1) and the overexpression classification (pcDNA3.1-PER2). Subsequent to finishing the treatment, cell pellets were acquired and went through two washes using cold phosphate-buffered saline (PBS). To investigate apoptosis, the cell pellet was rehydrated in 100 μL of 1× Binding Buffer to yield a uniform single-cell suspension. Subsequently, the suspension received 5 μL of the staining solutions, which were Annexin V-FITC and propidium iodide (PI). The resultant mixture was meticulously homogenized, shielded from photonic exposure, and allowed to incubate at ambient temperature for a duration of 10 min. Following this incubation period, 400 μL of 1× Binding Buffer was introduced to the cell suspension. The sample was gently mixed and analyzed by flow cytometry within 1 h to assess apoptosis levels.

### 4.9. Detection of Related Genes

To detect the expression of apoptosis-related genes, primers were designed using the NCBI Primer-BLAST tool. The target genes for this analysis included *BCL2*, *CASPASE3*, *CASPASE8*, *CASPASE9*, *p53*, *p21*, *BAX*, *CASPASE6* and *CASPASE7*.

### 4.10. Cellular Reactive Oxygen Species, (ROS) Detection

To measure intracellular levels of reactive oxygen species (ROS), we used a reactive oxygen species detection kit. The ROS detection kit was sourced from Thermo Fisher Scientific, with catalog number D399. Cells were seeded at a density of 2 × 10^5^ cells/mL in 6-well plates, with both control (pcDNA3.1) and overexpression (pcDNA3.1-*PER2*) groups prepared. After the designated treatment period, the medium was removed, and the cells were washed twice with PBS. Then, 1 mL of 10 μmol/L DCFH-DA working solution, freshly prepared and protected from light to prevent degradation, was added to each well. Cells were exposed to DCFH-DA solution at 37 °C for 20 min. Post-incubation, the solution was extracted, and the cells underwent three washes with serum-free medium to remove excess dye. The fluorescence was then observed using a laser confocal microscope to assess ROS levels within the cells.

### 4.11. Detection of Lactate Dehydrogenase (LDH) Activity in Cells

To assess the enzymatic performance of lactate dehydrogenase (LDH), cellular samples were placed into 6-well plates at a concentration of 3 × 10^5^ cells in each well, including three replicates’ samples for every experimental group. The experimental groups included untransfected cells, pcDNA3.1-transfected cells (serving as the maximum enzyme activity control), and pcDNA3.1-*PER2*-transfected cells. These plates were incubated at 37 °C with 5% CO_2_ for the specific experimental time. One hour prior to the measurement phase, the cell culture plates were withdrawn from the incubator, and an LDH release reagent was introduced into the wells intended for the maximal enzyme activity control at a volume corresponding to 10% of the initial culture volume. The reagent was thoroughly mixed by pipetting, and the plates were returned to the incubator for continued reaction. Following the incubation period, 120 μL of the liquid from all wells was moved to a fresh 96-well plate, followed by the addition of 60 μL of LDH detection solution. The plate was subsequently covered to exclude light exposure and maintained at ambient temperature for a duration of 30 min.

### 4.12. Detection of Antioxidant Properties of Cells

To evaluate the antioxidant properties of the cells, we used an antioxidant detection kit from Nanjing Jiancheng Company, Nanjing, China. The procedure was performed according to the manufacturer’s instructions to ensure accurate and reliable results. Each step was carefully followed as outlined in the product guidelines to effectively assess the antioxidant capacity of the cells.

### 4.13. Statistical Analysis

FlowJo V10 software was used to interpret the flow cytometry results for apoptosis data. Cell cycle data were analyzed using ModFit LT software version 6.0 (https://www.vsh.com/products/mflt, accessed on 4 November 2024) to accurately assess cell cycle distributions. Statistical analysis and graphical representation of the data were performed using SPSS 25.0, Excel 2016, and GraphPad Prism software. These tools facilitated comprehensive statistical evaluation and the creation of informative charts and graphs to effectively present the research findings.

## 5. Conclusions

The circadian clock genes are highly expressed in cultured (REC) rumen epithelial cells. This study indicates that elevated levels of *PER2* within these cells significantly promote apoptosis while simultaneously inhibiting cellular proliferation. This effect is achieved through the upregulation of pro-apoptotic genes like CASP6 and BAX and the downregulation of the anti-apoptotic gene *BCL2*. The overexpression of *PER2* modulates the transcriptional activity of key genes associated with the cell cycle, highlighting its crucial role in balancing cell proliferation and apoptosis. These findings underscore the significant regulatory functions of *PER2* within cellular mechanisms and provide a strong foundation for further investigations into its underlying processes.

## Figures and Tables

**Figure 1 ijms-25-12428-f001:**
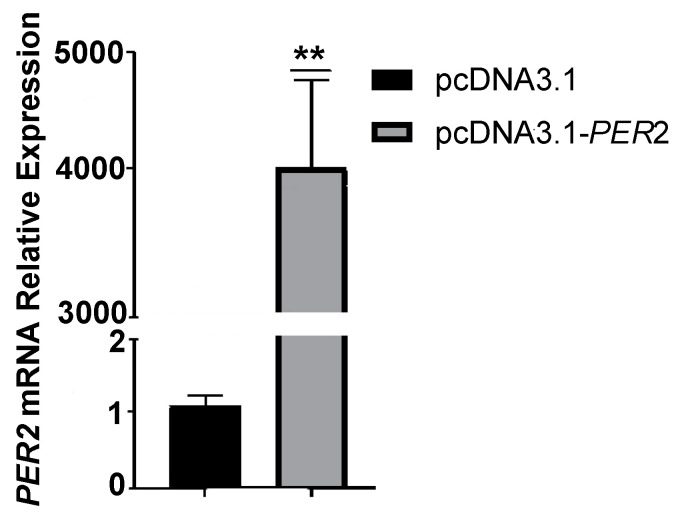
Relative expression of *PER2* gene in goat rumen epithelial cells. The data of each group were mean ± SEM (n = 3), ** showed significant difference (*p* ≤ 0.01).

**Figure 2 ijms-25-12428-f002:**
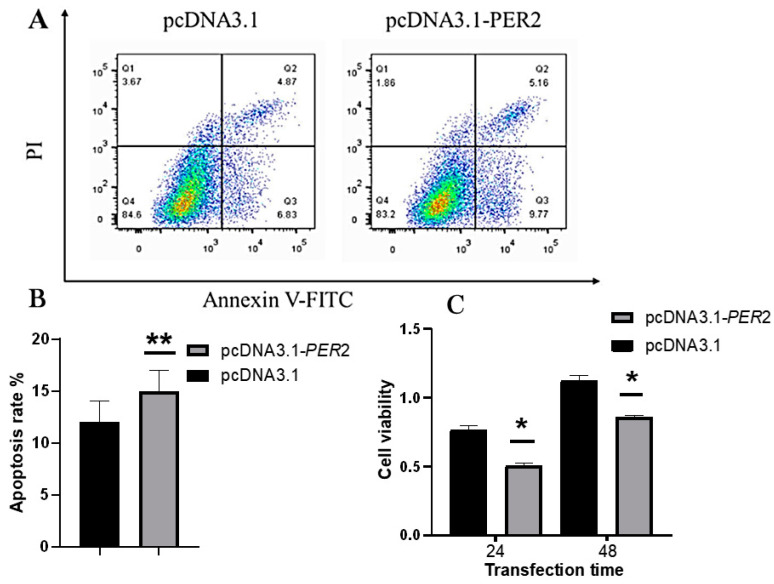
Influence of *PER2* overexpression on goat rumen epithelial cell proliferation and apoptosis. (**A**,**B**) Consequences of *PER2* overexpression on goat rumen epithelial cell apoptosis; (**C**) consequences of *PER2* overexpression on goat rumen epithelial cell proliferation. Each group’s data are represented as mean *±* SEM (n = 3), ** indicates a significant difference (*p* ≤ 0.01), * indicates a significant difference (*p* ≤ 0.05).

**Figure 3 ijms-25-12428-f003:**
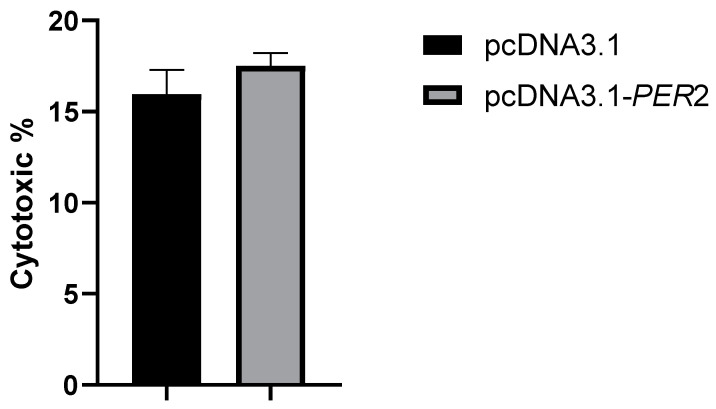
Effect of *PER2* overexpression on membrane damage in goat rumen epithelial cells.

**Figure 4 ijms-25-12428-f004:**
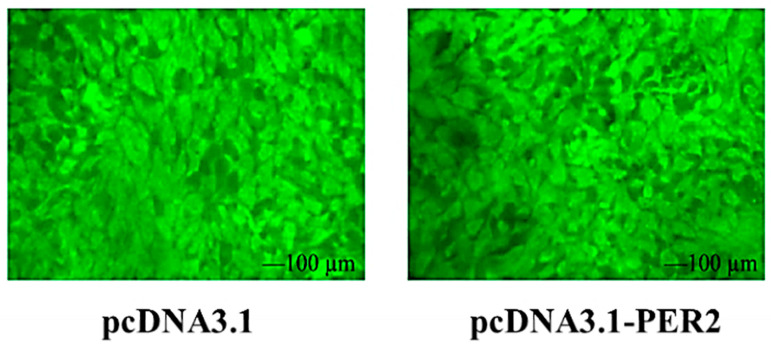
Effect of *PER2* overexpression on ROS levels in goat rumen epithelial cells.

**Figure 5 ijms-25-12428-f005:**
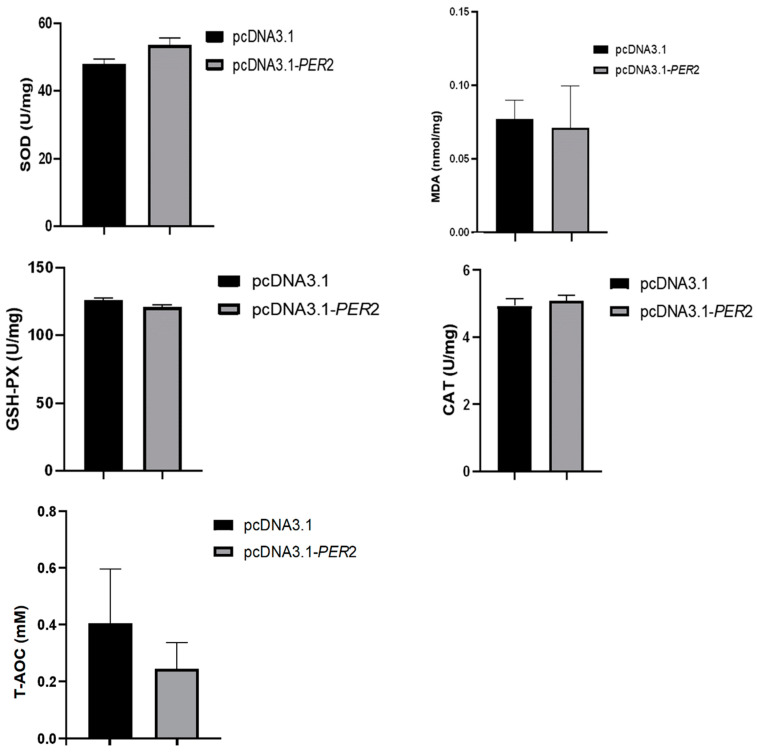
Effect of *PER2* overexpression on antioxidant markers in goat rumen epithelial cells.

**Figure 6 ijms-25-12428-f006:**
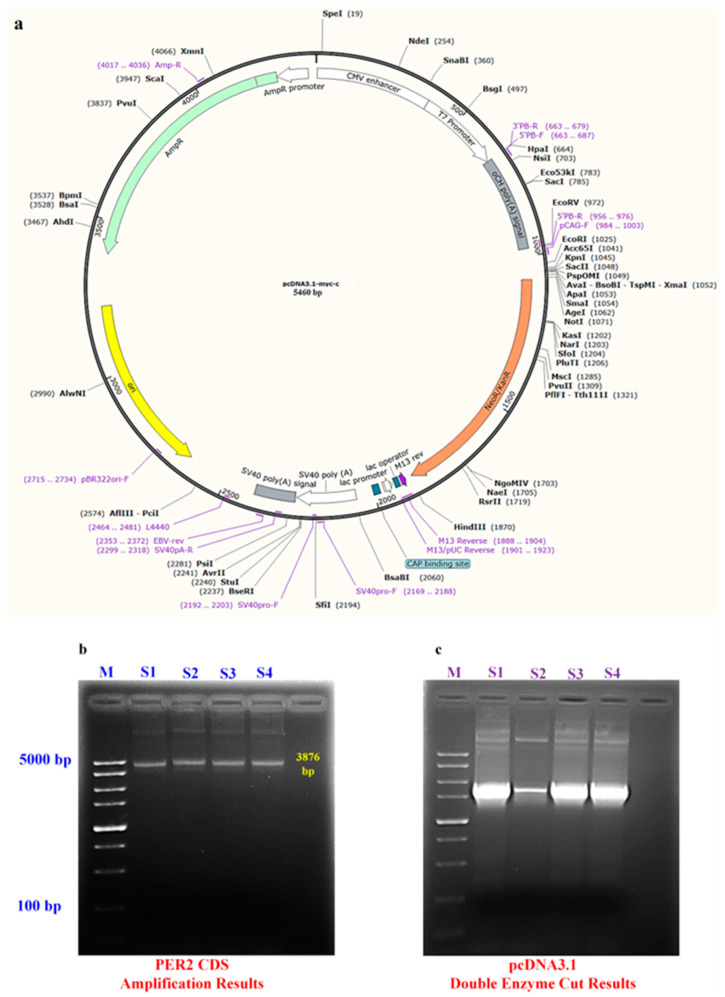
(**a**) Construction model of overexpression vector, (**b**) *PER2* CDS amplification results, and (**c**) pcDNA3.1 double enzyme cut results. M; 5 kb.

**Table 1 ijms-25-12428-t001:** Effect of *PER2* overexpression on the expression of genes related to the apoptosis pathway in goat rumen epithelial cells.

Gene	Constituencies		*p-*Value
	Control Group	Overexpression Group	
*BCL2*	1.00 ± 0.04 ^a^	0.83 ± 0.01 ^b^	<0.001
*CASPASE3*	1.00 ± 0.06	1.20 ± 0.42	0.488
*CASPASE8*	1.00 ± 0.03	3.28 ± 1.80	0.093
*CASPASE9*	1.00 ± 0.13	1.15 ± 0.25	0.347
*p53*	1.00 ± 0.02 ^b^	2.73 ± 0.80 ^a^	0.043
*p21*	1.00 ± 0.02 ^b^	2.87 ± 0.79 ^a^	0.015
*BAX*	1.00 ± 0.06 ^b^	1.68 ± 0.17 ^a^	0.003
*CASPASE6*	1.00 ± 0.02 ^b^	3.28 ± 0.85 ^a^	0.010
*CASPASE7*	1.00 ± 0.03	0.76 ± 0.20	0.109

Note: data with different superscript letters indicate a significant difference (*p* ≤ 0.05), while data with the same superscript letters indicate no significant difference (*p* > 0.05).

**Table 2 ijms-25-12428-t002:** Effect of *PER2* overexpression on cell cycle gene expression in goat rumen epithelial cells.

Gene	Constituencies		*p-*Value
	Control Group	Overexpression Group	
*CCND1*	1.02 ± 0.06 ^b^	2.22 ± 1.00 ^a^	0.035
*CCNE1*	1.00 ± 0.06	1.20 ± 0.42	0.520
*CCNB1*	1.01 ± 0.03	1.17 ± 0.18	0.062
*CDK1*	1.00 ± 0.13	1.58 ± 0.27	0.309
*CDK2*	1.02 ± 0.06	0.82 ± 0.32	0.135
*CDK4*	1.02 ± 0.04 ^a^	0.74 ± 0.17 ^b^	0.041
*WEE1*	1.01 ± 0.01 ^b^	1.58 ± 0.07 ^a^	0.033
*p21*	1.00 ± 0.02 ^b^	2.87 ± 0.79 ^a^	0.015
*p16*	1.01 ± 0.01 ^b^	1.47 ± 0.05 ^a^	0.039

Note: data with different letter superscripts mean a significant difference (*p* ≤ 0.05), while data with the same letter superscripts mean no significant difference (*p* > 0.05).

**Table 3 ijms-25-12428-t003:** Effect of *PER2* overexpression on the expression of inflammatory genes in goat rumen epithelial cells.

Gene	Constituencies		*p-*Value
	Control Group	Overexpression Group	
*TNF-α*	1.01 ± 0.03 ^a^	1.40 ± 0.10 ^b^	0.051
*TLR-4*	1.04 ± 0.06 ^a^	1.26 ± 0.22 ^b^	0.096
*IL-6*	1.02 ± 0.04 ^a^	1.35 ± 0.03 ^b^	0.736
*IL-1β*	1.01 ± 0.01 ^a^	1.11 ± 0.13 ^a^	0.791

Note: data with different letter superscripts mean a significant difference (*p* ≤ 0.05), while data with the same letter superscripts mean no significant difference (*p* > 0.05).

**Table 4 ijms-25-12428-t004:** Impact of *PER2* overexpression on VFA absorption-related gene expression in goat rumen epithelial cells.

Gene	Constituencies		*p*-Value
	Control Group	Overexpression Group	
*AE2*	1.03 ± 0.13 ^a^	0.79 ± 0.06 ^b^	0.003
*NHE1*	1.18 ± 0.18 ^a^	0.59 ± 0.11 ^b^	0.008
*NHE3*	1.05 ± 0.31	1.11 ± 0.20	0.317
*NA/K ATPase*	1.00 ± 0.21	1.77 ± 0.48	0.501
*VH+ ATPase*	1.00 ± 0.17 ^b^	2.40 ± 0.27 ^a^	0.029
*MCT1*	1.00 ± 0.05 ^a^	0.74 ± 0.05 ^b^	0.021
*MCT4*	1.00 ± 0.03 ^a^	0.75 ± 0.05 ^b^	0.021
*PAT1*	1.09 ± 0.02 ^b^	1.78 ± 0.23 ^a^	0.006

Note: data with different letter superscripts mean a significant difference (*p* ≤ 0.05), while data with the same letter superscripts mean no significant difference (*p* > 0.05).

## Data Availability

All data generated or analyzed during this study are included in this manuscript and its information files.

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
