# Peer review of "Impact of Circadian Clock PER2 Gene Overexpression on Rumen Epithelial Cell Dynamics and VFA Transport Protein Expression"

_ijms, 2024, doi:10.3390/ijms252212428_

Round 1
Reviewer 1 Report
Comments and Suggestions for Authors
General comment
This study offers valuable insights into the role of PER2 gene in regulating apoptosis, cell proliferation, and transporter expression within goat rumen epithelial cells. The findings shed light on the molecular interactions that PER2 overexpression induces, highlighting its potential impact on rumen physiology. Generally, the paper is really good, the length of the manuscript is proper, most of the data presented are in a proper and scientific way. Here there are major and minor comments:
Major comments:
~ To enhance the manuscript's clarity and impact, I suggest revising the discussion to more thoroughly address the biological implications of the observed changes in gene expression and cell dynamics. Specifically, the discussion could benefit from an expanded the findings of other reliable published studies.
~ Additionally, consider integrating a comparison of these findings with existing literature on PER2's role in other species. Such a comparison could provide a more comprehensive understanding of how PER2 functions across different biological systems and reinforce the significance of your study within the field of circadian biology. This detailed examination within the discussion will better elucidate the potential physiological and metabolic consequences of PER2 overexpression, presenting a clear narrative that aligns with the study's objectives.
~ In the introduction section, I suggest authors to add a paragraph that briefly explains the critical role of the rumen in nutrient absorption and overall ruminant health. Highlight why studying the rumen is important for understanding ruminant physiology and how circadian regulation could impact these processes. This will establish the biological and practical relevance of the study, emphasizing the impact of circadian rhythms on a vital organ in ruminants.
~ Also, explicitly state the current gap in understanding the role of circadian clock genes in the rumen compared to other tissues such as the liver and mammary glands. Highlight why this gap merits investigation. By clearly identifying the knowledge gap, you provide a strong rationale for your research, underscoring its novelty and importance.
~ In the discussion section, consider how understanding PER2's role in the rumen might influence livestock management, dietary formulations, or enhance ruminant productivity.
~ In the methods section, aim to answer the following question simply and in detail;
~ How were the rumen epithelial cells (REC) isolated and prepared for the experiments?
~ What methods were employed to confirm the successful overexpression of the PER2 gene in the rumen epithelial cells?
~What specific assays or tests were conducted to assess the effects of PER2 overexpression on rumen epithelial cells?
Comments on the Quality of English LanguageThe manuscript is well written, however, the author need to critically revise the manuscript to avoid any mistake
Author Response
Dear Reviewer, We sincerely thank you for the valuable feedback and constructive insights. Your comments have been instrumental in enhancing the clarity, rigor, and overall quality of our manuscript. We appreciate your time and expertise in helping us improve our work. Please see the attachment file (point-by-point response for each comment for both reviewers). Thanks.

Reviewer 2 Report
Comments and Suggestions for Authors
Ali et al. provide an in-depth analysis of the impact of PER2 gene overexpression on goat rumen epithelial cells, examining its effects on cell proliferation, apoptosis, and VFA transporter expression. However, significant revisions are needed to improve data presentation and clarify the methodology. The authors should also carefully proofread the manuscript.
- Figure quality and details: The figures are of low quality, and important information is missing. Individual plots should be shown where relevant. For example, figures 1B and 1C lack ladder sizes, and the y-axis title in figure 3C should be changed to “cell viability” instead of “OD450.” Additionally, figure 5 should include DAPI staining for clearer visualization.
- Experimental details: Essential experimental details are missing, which limits reproducibility. For instance, the paper lacks information on the sex of the goats, the human age equivalent of a 1.5-year-old goat, and the source (company and catalog number) for ROS measurement reagents.
- Language and consistency: The manuscript contains multiple typographical errors and inconsistent formatting. For example, the use of uppercase and lowercase in terms like “p” and “P” as well as “p53” and “P53” is inconsistent. CO2 should be correctly formatted as COâ‚‚. An English editing service is recommended to improve readability and ensure consistent terminology.
- Choice of model: The study relies on a single cell line from goat rumen epithelium, which limits the generalizability of the findings. The authors should either justify this choice or consider using alternative models to validate the results across different cell types, which would help establish broader applicability.
5. Lack of Circadian Time-Course Data: The study focuses on PER2, a circadian gene, but only reports effects at fixed time points (24 and 48 hours post-transfection). Including a time-course analysis would help capture any rhythmic patterns in PER2’s impact on cell dynamics, which is crucial for circadian studies.
6. Limited Discussion of Mechanisms: The paper shows that PER2 overexpression affects apoptosis, cell cycle, and VFA transporter expression, but does not sufficiently discuss the molecular mechanisms. Including more on how PER2 might interact with pathways like p53 to drive these changes would provide valuable context and strengthen the study’s impact.
Comments on the Quality of English LanguageEnglish editing service is recommended to improve readability and ensure consistent terminology.
Author Response
Dear Reviewer, we are deeply grateful for your insightful feedback and constructive suggestions. Your thorough review has significantly contributed to improving the clarity and quality of our manuscript. We greatly value your expertise and the time you dedicated to helping us enhance our work. Thank you sincerely for your support. Please see the attachment file (point-by-point response for each comment for both reviewers). Thanks.

Round 2
Reviewer 2 Report
Comments and Suggestions for Authors
The authors responded to the reviewers' questions.
Comments on the Quality of English LanguagePlease use an English editing service before accepting. There are still many typos. For example, in line 357, 'P21' should be 'p21'.